

# Micrococcal nuclease sequencing of porcine sperm suggests enriched co-location between retained histones and genomic regions related to semen quality and early embryo development

Marta Gòdia[1,2,*], Yu Lian[1,*], Marina Naval-Sanchez[3], Inma Ponte[4], Joan Enric Rodríguez-Gil[5], Armand Sanchez[1,6] and Alex Clop[1,7]

[1] Centre for Research in Agricultural Genomics CRAG (CSIC-IRTA-UAB-UB), Cerdanyola del Vallés, Catalonia, Spain
[2] Animal Breeding and Genomics, Wageningen University and Research, Wageninger, Netherlands
[3] Agriculture & Food, CSIRO, Brisbane, Queensland, Australia
[4] Biochemistry and Molecular Biology, Universitat Autònoma de Barcelona, Cerdanyola del Vallés, Catalonia, Spain
[5] Animal Medicine and Surgery, Universitat Autònoma de Barcelona, Cerdanyola del Vallés, Catalonia, Spain
[6] Animal and food sciences, Universitat Autònoma de Barcelona, Cerdanyola del Vallés, Catalonia, Spain
[7] Consejo Superior de Investigaciones Científicas, Barcelona, Catalonia, Spain
* These authors contributed equally to this work.

Corresponding authors
Marta Gòdia, marta.godia@wur.nl
Alex Clop, alex.clop@csic.es

## ABSTRACT

The mammalian spermatozoon has a unique chromatin structure in which the majority of histones are replaced by protamines during spermatogenesis and a small fraction of nucleosomes are retained at specific locations of the genome. The sperm's chromatin structure remains unresolved in most animal species, including the pig. However, mapping the genomic locations of retained nucleosomes in sperm could help understanding the molecular basis of both sperm development and function as well as embryo development. This information could then be useful to identify molecular markers for sperm quality and fertility traits. Here, micrococcal nuclease digestion coupled with high throughput sequencing was performed on pig sperm to map the genomic location of mono- and sub-nucleosomal chromatin fractions in relation to a set of diverse functional elements of the genome, some of which were related to semen quality and early embryogenesis. In particular, the investigated elements were promoters, the different sections of the gene body, coding and non-coding RNAs present in the pig sperm, potential transcription factor binding sites, genomic regions associated to semen quality traits and repeat elements. The analysis yielded 25,293 and 4,239 peaks in the mono- and sub-nucleosomal fractions, covering 0.3% and 0.02% of the porcine genome, respectively. A cross-species comparison revealed positional conservation of the nucleosome retention in sperm between the pig data and a human dataset that found nucleosome enrichment in genomic regions of importance in development. Both gene ontology analysis of the genes mapping nearby the mono-nucleosomal peaks and the identification of putative transcription factor binding motifs within the mono- and

the sub- nucleosomal peaks showed enrichment for processes related to sperm function and embryo development. There was significant motif enrichment for Znf263, which in humans was suggested to be a key regulator of genes with paternal preferential expression during early embryogenesis. Moreover, enriched positional intersection was found in the genome between the mono-nucleosomal peaks and both the RNAs present in pig sperm and the RNAs related to sperm quality. There was no co-location between GWAS hits for semen quality in swine and the nucleosomal sites. Finally, the data evidenced depletion of mono-nucleosomes in long interspersed nuclear elements and enrichment of sub-nucleosomes in short interspersed repeat elements.These results suggest that retained nucleosomes in sperm could both mark regulatory elements or genes expressed during spermatogenesis linked to semen quality and fertility and act as transcriptional guides during early embryogenesis. The results of this study support the undertaking of ambitious research using a larger number of samples to robustly assess the positional relationship between histone retention in sperm and the reproductive ability of boars.

# INTRODUCTION

Current research is showing evidences that, in addition to the paternal genome, sperm also carries other molecular information to the zygote. This includes a wide repertoire of RNAs, proteins and epigenetic marks in the form of DNA methylation, retained nucleosomes and histone modifications that can play a role in spermatogenesis, fertility, early embryo development and even the transmission of inter-generational information (*Krawetz, 2005*; *Gòdia, Swanson & Krawetz, 2018*; *Spadafora, 2017*; *Hammoud et al., 2009*; *Oikawa et al., 2020*). During spermatogenesis, male germ cells undergo a series of profound morphological and functional changes that conclude in mature spermatozoa. In the last stages of spermatogenesis, when the cell is transcriptionally silent, genomic DNA (gDNA) is highly condensed to fit into the sperm head (*Ward & Coffey, 1991*) and ensure genomic integrity, early embryo development and ultimately, fertility (*Ward, 2010*). In mammals, the condensation of the spermatozoon chromatin occurs through the progressive replacement of histones by protamines. However, a small fraction of the sperm DNA remains organized in histones as shown in humans (5 to 15% of histones are retained in the sperm chromatin) (*Gatewood et al., 1987*; *Hammoud et al., 2009*; *Arpanahi et al., 2009*; *Brykczynska et al., 2010*; *Castillo et al., 2014*; *Samans et al., 2014*), cattle (13.4%) (*Samans et al., 2014*) and mice (1 to 2%) (*Balhorn, Gledhill & Wyrobek, 1977*; *Erkek et al., 2013*; *Carone et al., 2014*; *Johnson et al., 2016*). Several studies indicate that these nucleosomes are distributed along the genome following a programmatic pattern with preferential retention at gene promoters and developmental related loci (*Hammoud et al., 2009*; *Arpanahi et al., 2009*; *Brykczynska et al., 2010*; *Castillo et al., 2014*; *Jung et al., 2017*; *Yoshida et al., 2018*). However, these results are inconclusive because at least three other

publications reported that sperm nucleosomes are enriched at gene-poor regions or repeat elements (*Samans et al., 2014*; *Carone et al., 2014*; *Yamaguchi et al., 2018*) while they may even be depleted at promoters (*Carone et al., 2014*; *Samans et al., 2014*) and exons (*Samans et al., 2014*). The reasons underlying these discrepancies, which are not mutually exclusive, are unclear and could be attributable to technical differences in the protocols used in each study. Carone and colleagues found that nucleosomes differ in their resistance to micrococcal nuclease (MNase) digestion and that the most fragile nucleosomes tended to map near repeat elements while nucleosomes at promoters were more stable (*Carone et al., 2014*). Thus, protocols using large amounts of MNase would artifactually result in nucleosomal enrichment at promoters (*Carone et al., 2014*). However, other studies, which used alternative protocols not based on MNase and that claimed to have unbiasedly characterized all nucleosomes present in the sperm chromatin, found enriched nucleosomes at promoters or gene bodies (*Johnson et al., 2016*; *Jung et al., 2017*; *Yoshida et al., 2018*). Another study based on nucleoplasmin instead of MNase to solubilize the sperm's chromatin concluded enrichment at both, promoters and gene bodies as well as gene deserts (*Yamaguchi et al., 2018*). They hypothesized that MNase based protocols do not retain all nucleosomes (*Yamaguchi et al., 2018*).

Nucleosome positioning in the genome and chromatin accessibility is critical in the regulation of gene expression and the alteration of this epigenomic architecture has been linked to multiple phenotypes in a range of tissues and cell types (*Lai & Pugh, 2017*). Nucleosomes in sperm may either be leftovers of gene expression during spermatogenesis or provide transcriptional instructions upon fertilization for gamete recognition and early embryo development. *Jung et al. (2017)* found that a proportion of histone modifications in gene promoters of the mouse sperm chromatin, recapitulates the chromatin structure and transcriptional activity of late spermatids. Also, a wealth of studies reported that at least a proportion of histone modifications in sperm is maintained (*van der Heijden et al., 2006*; *Lismer et al., 2020*) and even that nucleosome positioning (*Ihara et al., 2014*) and the location of histone modifications (*Oikawa et al., 2020*) in sperm are related to gene expression in early embryos. Thus, this information could be of great value to shed light into the catalog of genomic regions and genes related to sperm biology and early embryo development and thus help identifying elusive molecular markers for traits related to sperm quality and fertility.

In livestock, the architecture of the sperm chromatin at the genomic level has only been investigated in cattle (*Samans et al., 2014*). In pigs (*Sus scrofa*), research has been carried to study the sperm RNA (*Gòdia et al., 2019*; *Gòdia et al., 2020a*; *Ablondi et al., 2021*) and protein (*Mendonça et al., 2017*) populations as well as the DNA methylation patterns (*Khezri et al., 2019*; *Pértille et al., 2021*), but the sperm's chromatin structure remains unexplored.

This study is sustained by the hypothesis that nucleosome retention in the pig sperm is not stochastic and owes to reminiscences of the spermatogenesis program and future information for the embryogenic program. Our objective was to map the retained nucleosomes in a pool of two pig sperm samples and evaluate their relative location in relation to different genomic features, some of which were associated to sperm biology,

semen quality and fertility. To fulfil this purpose, pig spermatozoa were digested with MNase and the resulting nucleosome-associated DNAs were subjected to high throughput sequencing. The mono-nucleosomal (MN) and sub-nucleosomal (SN) chromatin fractions were characterized and their correlation with sperm RNA levels and co-location with genomic regions associated to semen traits was assessed.

## MATERIALS AND METHODS

### Sample collection

Two ejaculates, each from a different healthy Pietrain boar, MN_1 (16 months of age) and MN_2 (9 months old), from two artificial insemination centres were obtained by specialists during their routine sample collection using the gloved hand method (*King & Macpherson, 1973*) and were immediately diluted (1:2) in commercial extender. Both ejaculates showed good semen quality as evidenced by the evaluation of several sperm parameters. The percentages of sperm cell viability, structurally altered acrosomes and morphological abnormalities were measured by staining the samples with the eosin-nigrosin technique after 5 min incubation at 37°C. These analyses showed 94.3% and 92.2% of viable cells for sample MN_1 and MN_2 respectively; 96.7% and 96.3% of cells with normal acrosomes, respectively; 2.2% and 1.1%, 3.1% and 1.1% and 2.2% and 4.6%, of head, neck and tail abnormalities, respectively. Semen samples were purified with BoviPure$^{TM}$ (Nidacon, Moelndal, Sweden) to remove potentially present somatic cells and processed as described in *Gòdia et al. (2018)*. Briefly, a variable volume of sperm according to its concentration, with a maximum of 1 billion cells and not exceeding 11 mL, were layered over 3 mL of BoviPure$^{TM}$ diluted to a final ratio of 60% (v/v) with BoviDilute$^{TM}$ (Nidacon, Moelndal, Sweden) and centrifuged at 300 × g for 20 min at 20 °C with slow acceleration and deceleration rates. The resulting cell pellet was washed with PBS and resuspended in 1 mL of RNase-free PBS and further pelleted by centrifugation. The pellets were stored at −80 °C until further use.

### Micrococcal nuclease digestion, library construction and sequencing

Chromatin digestion was performed as previously described (*Zalenskaya, Bradbury & Zalensky, 2000*) with minor modifications. Forty million purified spermatozoa cells were pelleted by centrifugation (3,500 rpm for 10 min) and washed twice with 1 mL of 1x phosphate buffered saline (PBS), and 1 mM phenylmethylsulfonyl fluoride (PMSF). Pelleted cells were resuspended with 1 mL of 0.1% Lysolecithin (dissolved in 1x PBS and 1 mM PMSF), incubated on ice for 10 min and centrifuged (3,500 rpm for 10 min). The sperm head was decondensed in 1 mL of 1x PBS, 1 mM PMSF with 10mM Dithiothreitol (DTT) and incubated on ice for 12 min. After centrifugation (2,000 rpm for 10 min), nuclei were washed with 1 mL of 1x PBS and 2mM DTT and centrifuged again (3,500 rpm for 10 min). The pellet was resuspended in 100 µl of 1x PBS, 1 mM DTT and CaCl$_2$ was added to a final concentration of 0.6 mM. At this time, 5 U of MNase (Sigma-Aldrich) were added and the samples were incubated for 7 min at 37 °C. Digestion was stopped by the addition of ethylene glycol-bis (β-aminoethyl ether)-N,N,N′,N′-tetraacetic acid (EGTA) to a concentration of 5 mM. The resulting digested fraction corresponding to

nucleosome-bound chromatin (soluble) was separated from the undigested protamine-bound fraction (pellet) by centrifugation at 20,000 rpm for 10 min. An aliquot of the digested DNA fraction (which included both the MN and SN fractions) was treated with sodium dodecyl sulfate (SDS) and proteinase K and evaluated in a 1.5% agarose 1x tris-acetate-EDTA (TAE) gel electrophoresis. Another aliquot of the nuclease digested DNA was used to extract DNA with a phenol-chloroform based protocol and was directly used for library prep after purification with the Agencourt AMPure XP beads (Beckman Coulter, Brea, CA, USA). The MN and SN fractions were separated after sequencing and read mapping, using bioinformatics tools as described below. Purified DNA was subjected to quality control including quantification with the Qubit$^{TM}$ DNA HS Assay kit (Invitrogen, Waltham, MA, USA) and Nanodrop (Thermo Fisher Scientific, Waltham, MA, USA) as well as size and concentration assessment with a 2100 Bioanalyzer and the Agilent DNA 1000 Kit (Agilent Technologies, Santa Clara, CA, USA). gDNA from the two sperm samples was also extracted as done by Hammoud and colleagues (*Hammoud et al., 2009*). Briefly, sperm cells were thawed and pelleted at 10,000 × g for 1 min. The supernatant was removed and the pellet was resuspended in 1 mL Lysis Reagent (1% Triton X-100, 5 mM MgCl2, 320 mM Sucrose and 10 mM Tris pH 7.5). After centrifugation (20,000 × g for 5 min), cells were resuspended again with 0.5mL Lysis Reagent two times, centrifuged and the supernatant was discarded. The sperm pellet was dispersed with 305.5μl of Enzyme Master Mix (1.25 mM MgCl$_2$, 1.25 mM Deferoxamine Mesylate, 12.5 mM Tris pH 8.0). Then, 1.6 μl RNase A-20 mg/mL -, 1.6 μl of 1 M DTT and 6.5 μl of 5M NaCl was added to each sample. This cell suspension was kept at 37 °C for 30 min. The cells were then dispersed with gentle vertexing and subsequently, 40 μl of 10% SDS and 20 μl of Proteinase K (20 mg/μl) were added. Samples were then incubated at 50 °C for 60 min. Following this step, 1 mL of 100% Isopropanol was added and the tubes were inverted several times until a gel-like DNA precipitate appeared. Then, 0.6 mL of NaI solution was added (40 mM Tris pH 8.5, 20 mM EDTA and 7.6 M NaI), and the tubes were inverted until DNA formed a white precipitate. The DNA pellet was washed three times with 70% EtOH with a final wash with 100% EtOH. The DNA pellet was solubilized with 300 μl of H$_2$O. The two resulting gDNA samples were pooled to be used as input. Pooled gDNA were sheared to obtain 100 bp long fragments using a Covaris S2 instrument (Covaris Inc, Woburn, MA, USA), according to the manufacturer's instructions. Sequencing libraries of the two MNase treated samples and the input gDNAs were prepared with the PrepX$^{TM}$ DNA Library Kit (Takara, Kusatsu, Japan). The libraries were used as a template to generate 50 bp paired end (PE) reads in an Illumina HiSeq2500 system.

## MNase-Seq data preprocessing and quality evaluation

Raw sequencing data was filtered to remove low quality bases, adaptor sequences and short reads (<25 bp length) with Trimmomatic v.0.36 (*Bolger, Lohse & Usadel, 2014*). Trimmed reads were aligned to the porcine genome (Sscrofa11.1) with Bowtie2 v.2.4.1 (*Langmead & Salzberg, 2012*) fitting default parameters except "—very sensitive". Pearson correlation of the read coverages for genomic regions along the genome was calculated using the

multiBamSummary file based on the two MNase samples with the tool deepTools v.3.3.2 (*Ramírez et al., 2016*) with options: "plotCorrelation—removeOutliers–skipZeros". Since the correlation between the samples was very high, the reads from the two replicates (MN_1: replicate A; MN_2: replicate B) were merged and processed as a pool with the aim to increase sequencing depth and peak detection power and accuracy. Then, the MN and SN fractions were bioinformatically separated based on the genomic distance between the two paired reads. The reads from the SN fraction, with a length slightly below 100 bp, were extracted using the parameter "–maxins 110", which specifies that the mapped paired-end reads should not exceed 110 bp. To obtain the reads from the MN fraction, with a length around 150 bp, the parameters "—minins 111" and "—maxins 1000" were applied. Subsequently, duplicate reads of these fractions and the input sample were removed with Picard-Tools MarkDuplicates v.1.56 (http://picard.sourceforge.net). Finally, for both fractions, peak calling was performed with MACS2 v.2.1.0 (*Zhang et al., 2008*) with "-q 0.05 -B -g hs" and compared against the input sample. Visualization of the genomic heatmaps from the MNase-Seq signals using transcription start sites (TSS) as reference points was carried with the deepTools (*Ramírez et al., 2016*) computeMatrix tool with parameters: "reference-point -b 500 -a 500 –skipZeros" and plotHeatmap tool using "—refPointLabel 'TSS'".

## Peak location relative to gene annotation

Peaks were categorized by their position in relation to gene features as annotated in Ensembl (v96) using BEDtools v.2.29.2 (*Quinlan & Hall, 2010*) intersect. The peaks were classified as mapping to TSSs, promoters, overlapping 5′ untranslated region (UTR), 3′ UTR, exonic, intronic and intergenic as carried in *Halstead et al. (2020)*.

In order to determine any potential positional preference or avoidance of the peaks within the gene features, 1,000 iterative permutations of the peak genomic locations with the same number of peaks following the same peak width distribution but with randomized genomic locations, were carried using BEDtools (*Quinlan & Hall, 2010*) shuffle. Subsequently, genomic features were assigned to the randomized peaks based on their genomic position within these features. These results were subsequently compared against our results on real data using a permutation test. This test calculates the proportion of permutations in which the number of peaks mapping to a specific feature deviates from the mean of the distribution more than the number of overlaps observed in the real data. In other words, it determines whether the peaks in the permutations are located more towards the tails of the distribution compared to the real data. The fold change (FC) was also measured. FC was calculated by dividing the number of overlaps between the real MN or SN peaks and each of the genomic features by the equivalent mean of the distribution of the number of overlaps obtained with the shuffled data. Comparisons resulting in *P-value* < 0.001 and FC ≥ 1.5 or FC ≤ 0.66 were classified as enriched or depleted, respectively.

## Gene ontology analysis of the genes near MNase peaks

The genes mapping within or less than 5 kbp apart from the identified peaks were extracted from Ensembl v96 with BEDtools (*Quinlan & Hall, 2010*) closestBed and were

used for Gene Ontology (GO) enrichment analysis. GO was carried out with Cytoscape v.3.8.2 plugin ClueGO v.2.5.7 (*Bindea et al., 2009*) using the Cytoscape's porcine dataset and default settings. Only the significant Bonferroni corrected *P*-values were considered.

## Motif enrichment analysis at the MN and SN underlying sequences

Motif enrichment analysis was carried out with the software HOMER v.4.10.0 (*Heinz et al., 2010*) findMotifsGenome.pl function with default parameters. These default parameters include: (i) random selection of the genome's background regions to compare against the input sequence data and using the background frequency of nucleotides in the human genome, hg19; (ii) the default motif length for the search is 8-12 nucleotides long; (iii) motif search on both the forward and reverse strands of the input sequences.

## Positional conservation of chromatin-associated DNA with human and cattle sperm

A comparable human sperm MNase-Seq dataset that sequenced the mono-nucleosomal fraction (GSE15690) from *Hammoud et al. (2009)*, obtained from a pool of four donors was downloaded from the Gene Expression Omnibus database. The genomic coordinates for the human sperm MN peaks were liftover to Sscrofa11.1 using the UCSC liftover tool (*Kuhn, Haussler & Kent, 2013*) with default parameters, except for the parameter "-minMatch = 0.1". To evaluate enrichment, the genomic locations of the porcine MNase peaks were randomized 1,000 times using BEDtools (*Quinlan & Hall, 2010*) shuffle and overlapped to the MNase peaks of human or cattle using BEDtools v.2.29.2 (*Quinlan & Hall, 2010*) intersect. Thereafter, the results were contrasted to the overlap of the real data with the permutation test that calculates the proportion of permutations in which the number of overlaped peaks lies further toward the tails of the distribution compared to the observed value in the real data. FC was also calculated by dividing the number of overlaps between the real MN or SN peaks and human MNase peaks by the mean of the distribution of the number of overlaps obtained from the shuffled data. To determine co-location enrichment or depletion, a threshold of *P*-value < 0.001 and FC ≥ 1.5 or FC ≤ 0.66, was set to determine enrichment and depletion, respectively.

The same approach was used for two cattle (GSE47843) comparable MNase-Seq datasets that sequenced the mono-nucleosomal fractions from two bulls (samples s_1_1_bovine and s_5_1_bovine) from the Holstein breed (*Samans et al., 2014*). Genome coordinates from bull (busTau7) genomes were first liftover to hg19, and then from hg19 to Sscr11.

## Integration with other -omics data

Although mature sperm is assumed to be transcriptionally inactive, RNAs in sperm may both reflect preceding events in transcriptionally active spermatogenic precursor cells during male germ cell development and have a role in early development after fertilization (*Gòdia, Swanson & Krawetz, 2018*; *Ostermeier et al., 2004*). Thus, the existence of a relationship between the location of retained nucleosomes in sperm and the sperm transcriptome profile was hypothesized. To test this hypothesis, sperm transcriptome data
from 40 healthy Pietrain boars belonging to commercial artificial insemination farms, including 40 total and 35 small RNA-seq datasets previously generated by our group (NCBI's BioProject PRJNA520978) was used. These datasets provided a list of 4,120 protein coding RNAs (*Gòdia et al., 2020b*), 1,574 circular RNAs (circRNAs) mapping in pig chromosomes (*Gòdia et al., 2020a*) and 6,729 PIWI-interacting RNAs (piRNAs) (*Ablondi et al., 2021*). The averaged RNA abundances from all the samples were used for further analysis. Protein coding genes were classified according to their RNA abundances measured in Fragments per Kilobase per Million mapped reads (FPKM) as (i) absent (<1 FPKM); (ii) low abundance (≥1 to <10 FPKM); (ii) intermediate abundance (≥10 to <100 FPKM) and (iii) high abundance (≥100 FPKM). Then, the existence of positional co-location enrichment between each of these gene fractions and the MNase peaks was assessed using the Fisher Exact Test (two-tailed). Any distance below 5 kb between an RNA and a MN or SN peak was considered as co-location. Moreover, the RNA abundance of the genes that co-located with MN and SN peaks with the RNA levels of the whole set of genes annotated in the pig genome was compared by employing the Kruskal-Wallis test using RNA abundances stabilized with the log2 transformation.

The positional enrichment of circRNAs or piRNAs at MNase peaks was also studied. First, the genomic regions of the MN and SN peaks were iteratively randomized 1,000 times using BEDtools (*Quinlan & Hall, 2010*) shuffle. The overlap in the real data and the permutations with the circRNAs or piRNAs was determined with BEDtools v.2.29.2 (*Quinlan & Hall, 2010*) intersect. Then, the enrichment of the genomic coordinates of the real MNase peaks at piRNA and circRNAs was assessed by comparison with the overlap of the randomized MNase peak locations using the permutation test. The FC was also calculated by dividing the number of overlaps between the real MN or SN peaks and circRNAs or piRNAs by the mean of the distribution of the number of overlaps obtained with the shuffled data. A threshold of *P*-value < 0.001 and FC ≥ 1.5 (enrichment) or FC ≤ 0.66 (depletion), respectively, was set to consider co-location enrichment or depletion.

MNase profiles were also evaluated against the genomic regions showing genetic association with sperm quality traits in Pietrain pigs. These associations were identified through a Genome-Wide Association Study (GWAS) conducted by our group, which included the two samples used in this study (*Gòdia et al., 2020b*). This GWAS provided 18 regions showing genetic association with the percentage of sperm cells with head abnormalities (seven regions), neck abnormalities (four regions), percentage of abnormal acrosomes after 5 min incubation at 37 °C (two regions), the ratio of abnormal acrosomes after 5 min and 90 min of incubation (one region), percentage of motile spermatozoa after 5 min incubation (two regions) and after 90 min incubation (two regions, one shared with motility after 5 min incubation) and the percentage of sperm cells with proximal droplets (one region). These regions spanned in total, 16.5 Mbp (0.7%) of the porcine autosomal genome (*Gòdia et al., 2020b*). Again, to evaluate whether MNase peaks were enriched at GWAS hits, the genomic regions of the MN and SN peaks were iteratively randomized 1,000 times using BEDtools (*Quinlan & Hall, 2010*) shuffle and the overlap of both, the real observed data and the 1,000 randomized files with the GWAS hits was evaluated with BEDtools v.2.29.2 (*Quinlan & Hall, 2010*) intersect. This randomized overlap was

compared to the overlap of the real data using the permutation test. The FC was also calculated by dividing the number of overlaps between the real MN or SN peaks and the GWAS hits by the mean of the distribution of the number of overlaps obtained with the shuffled data. A threshold of $P$-value < 0.001 and FC = 1.5 (enrichment) or FC = 0.66 (depletion), respectively, was set to consider co-location enrichment or depletion.

### Peak location relative to repeat elements

The list of repeat elements in the pig genome from RepeatMasker was downloaded from the UCSC genome browser. The same approach as described for the other genomic features was followed to evaluate enrichment or depletion of MNase peaks at the whole set of repeat elements, long interspersed nuclear elements (LINEs) or short interspersed nuclear elements (SINEs). The genomic locations of the MN and the SN peaks were randomized 1,000 times and the overlap with the different sets of repeat elements was assessed with BEDtools v.2.29.2 (*Quinlan & Hall, 2010*) intersect. A permutation test was carried to evaluate enrichment ($P$-value < 0.001; FC ≥ 1.5) or depletion ($P$-value < 0.001; FC ≤ 0.66) of MN and SN peaks at repeat elements.

All scripts, including R, bash and software parameters used are available on Figshare: https://doi.org/10.6084/m9.figshare.21997523.v1.

## RESULTS

### MNase digestion, library preparation and data preprocessing

The MNase protocol yielded 95.2 and 75.6 ng of DNA for sample 1 and sample 2, respectively. The extraction of gDNA provided 1227.6 ng/ul. The MNase treatment of the pig sperm samples generated ~147 bp fragments corresponding to the MN fraction and an additional ~100 bp fragment corresponding to the SN fraction (Fig. S1).

The sequencing of the libraries yielded more than 43 million PE reads for each MNase biological replicate and 41.8 million PE reads for the input sample (Table 1). In average, 90.2% of the MNase reads and 92.7% for the input reads mapped to the porcine genome (Sscrofa11.1). The genome-wide profiles of MNase sensitivity of the two samples showed high correlation (Pearson R = 0.87) (Fig. S2) and similar mapping statistics (Table 1). Thus, the sample pooling strategy carried by Hammoud and collaborators (*Hammoud et al., 2009*) was replicated and the two samples were pooled with the aim to increase read depth and peak call sensitivity and accuracy. The final number of PE reads in each, the MN and SN fractions, was 62.1 and 7.1 million, respectively (Table 1).

### Characterization of the MN and SN peaks

The analysis yielded 25,293 MN and 4,239 SN peaks (Tables S1 and S2). The MN peaks averaged 270 bp in width and covered 0.3% of the porcine genome. The SN peaks were 141 bp wide and covered 0.02% of the genome. Most MN (70.2%) and SN (94.5%) peaks were annotated in intronic and intergenic regions (Fig. 1). MN peaks were significantly enriched at promoters ($P$-value < 0.001, FC = 5.05), TSS ($P$-value < 0.001, FC = 11.26) (Fig. 2), 5′-UTR ($P$-value < 0.001, FC = 12.20), coding sequences ($P$-value < 0.001, FC = 4.16) and 3′-UTR ($P$-value < 0.001, FC = 1.56) (Table 2 and Fig. S3). This result was not replicated in

**Table 1 MNase-Seq sequencing and pre-processing metrics for the input and the two MNase biological replicates.**

|  | Input DNA | MNase replicate A | MNase replicate B |
|---|---|---|---|
| Sequencing reads (PE) | 41,805,617 | 44,372,118 | 43,267,605 |
| Reads mapped to Sscrofa11.1 | 38,332,480 | 39,906,605 | 38,303,527 |
| % mapping Sscrofa11.1 | 92.7% | 91.4% | 89.6% |
| % duplicates | 23.4% | 9.3% | 8.4% |
| Number of reads in the MN fraction (PE) |  | 62,133,428 |  |
| Number of reads in the SN fraction (PE) |  | 7,139,475 |  |

**Note:**
PE, paired end reads.

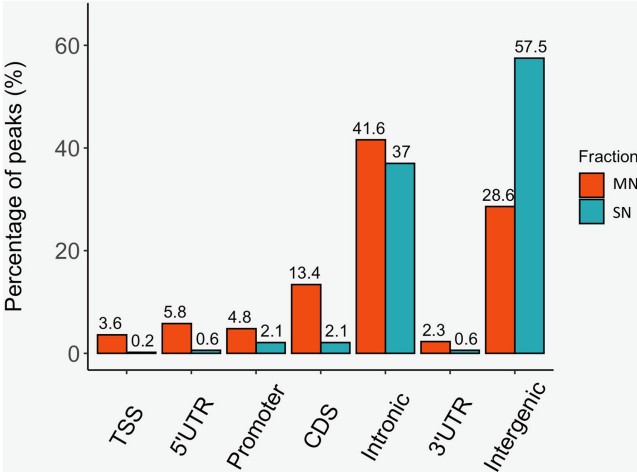

**Figure 1 Distribution of the MN and SN peaks relative to gene features in the pig genome.** Peaks were classified according to their co-location with gene features as, transcription start site (TSS), 5′ untranslated region (5′UTR), 3′UTR, promoter, coding sequence (CDS), intronic and intergenic.

the SN peaks, which showed mild depletion at 3'-UTR sites ($P$-value < 0.001, FC = 0.47) (Table 2 and Fig. S3). A total of 9,128 and 1,688 genes overlapped or localized less than 5 kbp apart from a MN and a SN peak, respectively (Table S3). The two MN and SN fractions co-existed in 1,145 genes (Table S3), which corresponds to 75% of the SN-associated genes.

To get an idea of the potential function of the nucleosome associated DNA, a GO analysis was conducted on the genes that overlapped with or were located within 5 kb of these peaks. For the MN peaks, the most significantly enriched terms were related to chemical perception, development (including the nervous system, multicellular organism, tissue, embryo, *etc.*) and cell and organ morphogenesis (Table S4). The genes located in the SN fractions were enriched for less GO terms and showed weaker significances and included nervous system process, cell projection, regulation of GTPase activity, organelle organization and Golgi vesicle transport (Table S5). To delve further into the potential functions of the nucleosome retention in sperm, the genomic sequences underlying the

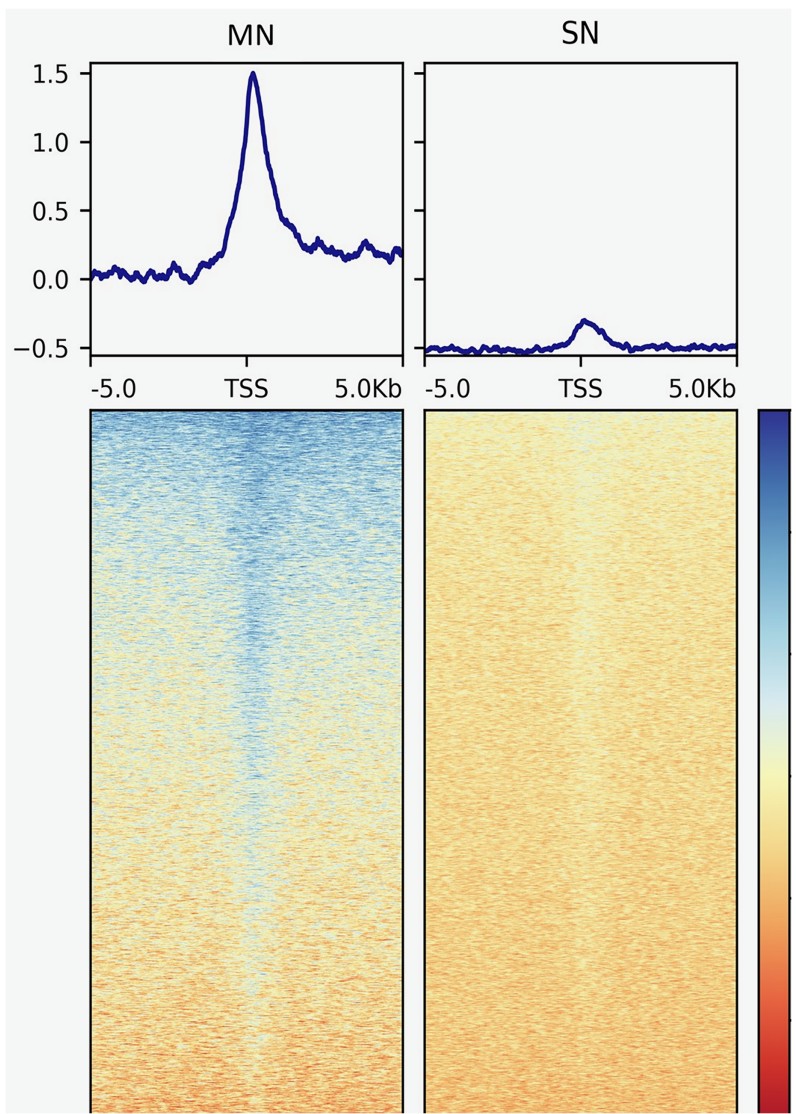

**Figure 2** **Genomic heatmaps depicting the normalized MNase-Seq signal centered at TSS for the MN and the SN peaks.** The x axis shows the genomic location relative to the TSS. The y axis indicates the MNase-Seq signal intensity.

peak regions were subjected to sequence motif enrichment analysis. Enrichment for motifs from 55 plant and 57 animal (mostly mammalian: mouse and human) transcription factors (TFs) was identified in the MN peaks (Table S6). The animal motifs involved 15 TFs from the bHLH class (MyoD, MyoG, Myf5, NeuroG2, Olig2, Tcf21, Twist2, *etc.*) and other TFs related to embryo development and implantation (Atf4, CHOP, Erra, EBF1, E2F2, Foxh1, HOXA1, HOXA2, MAFK, Pax8, IRF1, IRF3, PR, Rfx1, RORg, RUNX2, Zic, Zic3). They also included TFs related to spermatogenesis or sperm function which have also been related to embryo development such as CUX1, CTCFL, PAX5 and Smad2. Finally, this list also contained Znf263 and FOXA1, two TFs that provide a direct link between the paternal gamete and the embryo (Table S6). One study identified 514 genes with paternal preferential expression during human early embryo development and

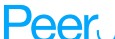

**Table 2 Summary of the results of the permutation tests to evaluate the location of the MN and SN peaks in relation to different genomic features.**

| | Source | Genomic feature | Real data | Min | Max | Mean | SE | *P*-val | Fold change |
|---|---|---|---|---|---|---|---|---|---|
| MN | Genome annotation | Promoter | 2,793 | 489 | 624 | 553.01 | 22.38 | <0.001 | 5.05 |
| | | TSS | 914 | 55 | 112 | 81.21 | 8.97 | <0.001 | 11.26 |
| | | 5'UTR | 2,070 | 128 | 205 | 169.74 | 12.99 | <0.001 | 12.20 |
| | | CDS | 3,386 | 734 | 891 | 814.82 | 28.07 | <0.001 | 4.16 |
| | | Intron | 10,527 | 9,956 | 10,458 | 10,168.04 | 79.43 | <0.001 | 1.04 |
| | | 3'UTR | 576 | 310 | 431 | 368.88 | 18.99 | <0.001 | 1.56 |
| | | Intergenic | 12,356 | 14,466 | 14,931 | 14,735.91 | 76.56 | <0.001 | 0.84 |
| | Data specific from pig sperm | circRNA | 3,198 | 2,679 | 3,023 | 2,847.75 | 48.15 | <0.001 | 1.12 |
| | | circRNA_corr | 14 | 5 | 27 | 15.03 | 3.82 | 0.46 | 0.93 |
| | | piRNAs | 65 | 4 | 29 | 14.54 | 3.85 | <0.001 | 4.47 |
| | | piRNAs_corr | 18 | 0 | 11 | 3.83 | 1.92 | <0.001 | 4.70 |
| | | GWAS | 189 | 123 | 221 | 171.05 | 12.98 | 0.09 | 1.10 |
| | RepeatMasker | RE | 12,666 | 16,080 | 16,597 | 16,341.85 | 74.27 | <0.001 | 0.78 |
| | | SINE | 5,892 | 6,968 | 7,421 | 7,182.70 | 68.15 | <0.001 | 0.82 |
| | | LINE | 3,826 | 7,651 | 8,085 | 7,889.53 | 72.58 | <0.001 | 0.48 |
| | Other species | Human | 5,395 | 163 | 265 | 215.71 | 15.23 | <0.001 | 25.01 |
| | | s_1_1_bovine | 7 | 3 | 34 | 17.08 | 4.31 | 0.009 | 0.41 |
| | | s_5_1_bovine | 16 | 12 | 44 | 26.75 | 5.11 | 0.014 | 0.60 |
| SN | Genome annotation | Promoter | 104 | 58 | 119 | 87.71 | 9.37 | <0.001 | 1.19 |
| | | TSS | 8 | 1 | 19 | 8.18 | 2.93 | 0.57 | 0.98 |
| | | 5'UTR | 27 | 7 | 41 | 22.06 | 4.76 | 0.18 | 1.22 |
| | | CDS | 88 | 73 | 145 | 103.62 | 10.06 | 0.06 | 0.85 |
| | | Intron | 1,568 | 1,575 | 1,806 | 1,693.45 | 32.20 | <0.001 | 0.93 |
| | | 3'UTR | 27 | 32 | 83 | 57.76 | 7.59 | <0.001 | 0.47 |
| | | Intergenic | 2,575 | 2,368 | 2,571 | 2,464.93 | 32.35 | <0.001 | 1.04 |
| | Data specific from pig sperm | circRNA | 460 | 400 | 542 | 477.26 | 20.71 | 0.21 | 0.96 |
| | | circRNA_corr | 4 | 0 | 8 | 2.45 | 1.60 | 0.25 | 1.63 |
| | | piRNAs | 5 | 0 | 9 | 1.59 | 1.29 | 0.02 | 3.15 |
| | | piRNAs_corr | 3 | 0 | 3 | 0.38 | 0.62 | 0.01 | 7.87 |
| | | GWAS | 24 | 14 | 48 | 28.92 | 5.48 | 0.22 | 0.83 |
| | RepeatMasker | RE | 3,559 | 2,381 | 2,579 | 2,485.37 | 32.50 | <0.001 | 1.43 |
| | | SINE | 1,817 | 890 | 1,062 | 979.63 | 27.83 | <0.001 | 1.85 |
| | | LINE | 874 | 1,065 | 1,250 | 1,163.82 | 29.09 | <0.001 | 0.75 |
| | Other species | Human | 88 | 17 | 50 | 31.73 | 5.55 | <0.001 | 2.77 |
| | | s_1_1_bovine | 1 | 0 | 9 | 2.68 | 1.61 | 0.249 | 0.37 |
| | | s_5_1_bovine | 1 | 0 | 11 | 3.64 | 1.92 | 0.112 | 0.28 |

**Note:**
Min, Max, Mean, minimum, maximum and mean number of overlaps for the 1,000 simulations, respectively. SE, standard error. TSS, transcription start site. 5'UTR, 5' untranslated region. CDS, coding sequence. 3'UTR, 3' untranslated region. circRNA_corr, circular RNAs which abundance level in sperm correlated to semen quality traits in swine as described by *Gòdia et al. (2020a)*. piRNA_corr, piRNAs which abundance level in sperm correlated to semen quality traits in swine as described by *Ablondi et al. (2021)*. GWAS, GWAS hits for semen quality traits in pigs as described by *Gòdia et al. (2020b)*. RE, repeat elements in the pig genome. SINE, short interspersed nuclear elements in swine. LINE, long interspersed nuclear element in swine.

proposed *ZNF263* as the strongest candidate in regulating the expression of these genes (*Leng et al., 2019*). The pig genes that map less than 500 bp apart from a MN peak harboring a Znf263 motif were identified, and their overlap with the 514 human genes showing paternal preferential expression in the early embryo was determined. 3,264 genes mapped less than 500 bp away from a MN peak with Znf263 binding motif in pig sperm. These corresponded to 3,288 human orthologs, 92 of which (18% of 514) also showed preferential paternal expression during human early embryo development (*Leng et al., 2019*) (Table S7). This concordance almost reached significance, with a *P*-value of 0.052 as calculated using the hypergeometric test. The SN peaks presented enriched motifs for 28 animal and 46 plant TFs. Thirteen of the animal motifs enriched in the SN peaks were also enriched in the MN fraction (Table S8). The common TFs included some of the bHLH class (NeuroG2, Olig2 and Twist2), as well as TFs involved in embryo development and implantation (Atf4, CHOP, ERRA, Foxh1, HOXA1, HOXA2, IRF1 and Zic).
The SN-specific enriched motifs corresponded to several TFs related to embryo development (NFX, Duxbl, PBX1, Pitx1, SF1, NFIL3, Gfi1b, HLF), meiosis (DMC1) and parent-of-origin expression driven by imprinting in embryo (Zfp57).

A comparable human sperm MNase-Seq dataset, which included pool data from four donors (*Hammoud et al., 2009*) and focused on MN peaks, was employed to assess the inter-species conservation of the genomic location of our MNase sites. Results in human were similar to these detected in our analysis in pig. 25,122 peaks were annotated in the human MN fraction, 20,641 of which were successfully converted to the pig genome coordinates. Twenty-six percent (5,395 peaks) of the human nucleosome associated sites overlapped with our porcine MN peaks. This is a highly significant co-location between the two species (*P*-value < 0.001, FC = 25.01), compared to randomization (Table 2 and Fig. S3). The two cattle samples contained 2,256 (s_1_1_bovine) and 8,446 (s_5_1_bovine) MN peaks. Of these, 1,011 (s_1_1_bovine) and 4,186 (s_5_1_bovine) were successfully liftover to pig and used to determine genomic overlap. Only 7 (s_1_1_bovine) and 16 (s_5_1_bovine) cattle MN peaks overlapped with the porcine MN peaks and no significant enrichment or depletion was observed.

To identify positional relationships between the nucleosome-associated regions and transcriptional activity, the locations of the MN and SN peaks were compared with the repertoire of RNAs that our group identified in porcine sperm (*Gòdia et al., 2020b*). A total of 12,125 protein coding genes were absent in sperm. On the contrary, 5,814, 3,521 and 598 genes were classified as being at low, moderate and high abundance in sperm, respectively. The location of the genes that are present in spermatozoa was, regardless of their abundance, highly enriched (low abundance: *P*-value = 1.8E−72, moderate abundance: *P*value = 1.5E−91, high abundance: *P*-value = 1.1E−11) within or near the MN peaks when compared to the catalog of absent genes (Table 3). Similarly, the SN peaks were also enriched for the low (*P*-value = 5.0E−17), moderate (*P*-value = 3.6E−47) and highly abundant (*P*value = 3.0E−5) genes, when compared to the set of absent genes (Table 3). In line with these results, the average RNA abundance of the genes mapping to both MN (*P*value = 4.9E−19) and SN (*P*-value = 4.3E−22) peaks was significantly higher than the average abundance of all the genes annotated in the pig genome (Fig. 3).

**Table 3 Distribution of the protein coding genes within the MN and SN fractions, according to their RNA abundance in sperm.**

| | | Genes within the MN peaks | | | Genes within the SN peaks | | |
|---|---|---|---|---|---|---|---|
| | | Less than 5 kb from a MN peak | More than 5 kb from a MN peak | *P*-value | Less than 5 kb from a SN peak | More than 5 kb from a SN peak | *P*-value |
| Gene's RNA levels in sperm | Not present | 4,083 | 8,042 | | 644 | 11,481 | |
| | Low abundance | 2,774 | 3,040 | 1.80E−72 | 504 | 5,310 | 5.00E−17 |
| | Intermediate abundance | 1,857 | 1,664 | 1.50E−91 | 455 | 3,066 | 3.60E−47 |
| | High abundance | 296 | 302 | 1.10E−11 | 60 | 538 | 3.00E−05 |

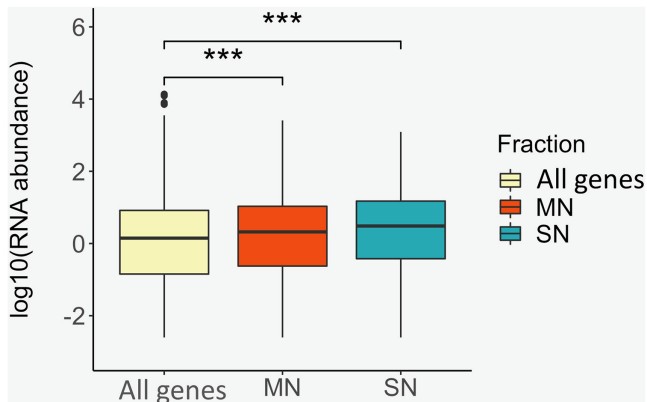

**Figure 3 Box plots showing the average RNA abundance of (i) all genes present in the genome (yellow), (ii) the genes present in the MN fraction (red) and (iii) the genes present in the SN fraction (blue).** ***: *P*-value ≤ 0.001.

Recently, our group annotated 6,729 sperm piRNAs in porcine sperm (*Ablondi et al., 2021*). The piRNA regions showed a significantly higher frequency of mapping within MN peaks (*P*-value < 0.001, FC = 4.47) and exhibited an overlap of 65 MN peaks (Table 2 and Fig. S3). A similar enriched co-location was also observed ((*P*-value < 0.001, FC = 4.70, overlap with 18 MN peaks) for the set of piRNA genomic regions involving 1,355 piRNAs (Table 2 and Fig. S3), which abundance in sperm was also found to correlate with various sperm phenotypes. These phenotypes include the percentage of motile sperm, the percentage of morphological abnormalities or the percentage of viable cells (*Ablondi et al., 2021*). No obvious enrichment or depletion of SN at piRNAs sites was observed (Table 2 and Fig. S3). No positive or negative co-location was observed between MN or SN peaks and the catalog of 1,574 sperm circRNAs and 148 circRNAs associated to sperm motility in swine (*Gòdia et al., 2020a*) (Table 2 and Fig. S3).

Our group recently published a GWAS for sperm quality traits in Pietrain boars (*Gòdia et al., 2020b*) which was used to evaluate the co-location of MNase sites with regions associated to these traits that could suggest a potential link between nucleosome retention

and sperm quality. No significant co-location between MN or SN and the GWAS hits was found (Table 2 and Fig. S3).

Finally, the MN peaks showed depletion at LINEs (*P*-value < 0.001, FC = 0.48), while the SN sites presented enrichment at SINEs (*P*-value < 0.001, FC = 1.85) (Table 2 and Fig. S3).

## DISCUSSION

This is a descriptive study that aimed at evaluating for the first time in swine, the tendency of the sperm's retained nucleosomes to co-map with different types of functional elements of the genome, which could be indicative of a non-random nucleosome retention. Echoing the results observed in other species (*Hammoud et al., 2009*; *Castillo et al., 2014*; *Samans et al., 2014*; *Carone et al., 2014*), the digestion resulted in two DNA bands (Fig. S1). The ~147 bp MN band corresponding to mono-nucleosomes consists of of DNA wrapped around the histone octamer comprising two copies of the core histones H2A, H2B, H3, and H4 (*Luger et al., 1997*). A SN band has been also detected in yeast (*Henikoff et al., 2011*), *Xenopus laevis* sperm (*Oikawa et al., 2020*), human sperm (*Castillo et al., 2014*), mouse sperm (*Carone et al., 2014*) and cattle sperm (*Samans et al., 2014*). The nature, composition and function of these short particles is still not well defined. Proteomic analysis of SN bands in *Xenopus* sperm revealed association with chromatin regulatory proteins (H3, H4, H1FX, CBX3, WDR5) and reduced amounts of H2A and H2B thereby leading to the hypothesis that SN are partially unwrapped nucleosomes that have lost one or the two H2A and H2B dimers (*Oikawa et al., 2020*). Other articles describe that at least in yeast and somatic cells, these partially unwrapped nucleosomes mark TFs to their cognate binding sequence, thus representing a novel signature of active chromatin as reviewed by *Brahma & Henikoff (2020)*. Another line of research suggests that SN particles in sperm corresponds to DNA sequences bound to CTCF (*Carone et al., 2014*), which is an important regulator of chromatin structure and transcriptional activity that has also been related to transgenerational inheritance from sperm to the offspring (*Jung et al., 2022*). In our study, there was no enrichment of CTCF binding sites in the SN peaks, even though enrichment of CTCFL, a CTCF paralog that is specific to the male germline, was observed in the MN fraction. Enrichment of putative binding sites for some TF was found, nearly half of which were also enriched in the MN peaks.

This, together with the opposite behavior observed between MN and SN regarding their location in relation to 3'UTRs and repeat elements (MN are depleted at LINEs and SN are enriched at SINEs) could be indicating that the SN fraction identified in our study may correspond to partially unwrapped nucleosomes generated during spermiogenesis or by the MNase digestion protocol.

The extraction of nucleosomal DNA was less efficient than in experiments carried in sperm from other species (*Hammoud et al., 2009*; *Brykczynska et al., 2010*; *Castillo et al., 2014*; *Samans et al., 2014*; *Erkek et al., 2013*), and it did not yield sufficient amount of DNA from the electrophoretic bands for high throughput sequencing. Nevertheless, the total amount of digested DNA was enough for sequencing and fragment size separation after read mapping using bioinformatics tools. To directly sequence the MN and SN fractions, future experiments will require to optimize the efficiency of the protocol or, less ideal, to

process a larger number of sperm cells to isolate sufficient DNA from each MNase electrophoretic band.

The nucleosome-associated DNA spanned ~0.3% of the porcine sperm's genome, which indicates a nearly fifteen, three and 45 -fold decrease in nucleosome retention when compared to what has been described in human (*Hammoud et al., 2009*; *Arpanahi et al., 2009*; *Gatewood et al., 1987*; *Brykczynska et al., 2010*; *Castillo et al., 2014*), mouse (*Balhorn, Gledhill & Wyrobek, 1977*; *Erkek et al., 2013*; *Johnson et al., 2016*), and cattle (*Samans et al., 2014*), respectively. This difference could have technical causes as previously described by Carone and co-authors regarding the existence of fragile and stable nucleosomes differing in their susceptibility to MNase digestion and the fact that the extent of MNase digestion can impact on the number of nucleosomes remaining in the sample (*Carone et al., 2014*). However, the existence of interspecies variability cannot be ruled out, which could be partially driven by differences in the protamine amino acid content resulting in changes on the extent or arrangement of the protamine disulfide bonding (*Perreault et al., 1988*).

An increasing body of evidence in other animal species points towards a programmatic nucleosome retention in the sperm genome (*Hammoud et al., 2009*; *Castillo et al., 2014*). In our study, the largest abundance of MN and SN peaks in intronic and intergenic regions (Fig. 1), is not unexpected since these gene features cover the vast majority of the genomic space. As a matter of fact, MN enrichment at promoters and exons (Table 2 and Fig. S3), provides the first indication that nucleosome retention in sperm may not be stochastic and may relate to the genome's functional activity. This hypothesis is further supported by the results obtained by comparing the genomic location of the MN peaks in porcine sperm with that in human; the gene annotation of the pig genome; the prediction of TF binding motifs and our own data on the pig sperm transcriptome.

The concordance with previous findings in other species regarding the predominant location of nucleosome peaks in gene promoters (*Hammoud et al., 2009*) and gene bodies (*Yamaguchi et al., 2018*) as well as the overlap with their human synthenic regions (*Hammoud et al., 2009*), indicates a degree of inter-species conservation in gene regulation, normally associated to the maintenance of important functions. Despite the significant overlap with the human MN map, still 73% of the human MN did not overlap to any of the porcine counterparts. This lack of full overlap could be attributed to both technical and biological reasons. As previously mentioned, the extend of MNase digestion could lead to differences in the MN profiles obtained by MNase-Seq (*Carone et al., 2014*). Furthermore, the possibility that porcine and human sperm have different sensitivities to the MNase digestion cannot be ruled out. The biological factors that could lead to the inter-species differences in the MN retention include the existence of a certain degree of random retention of nucleosomes due to their inefficient removal during spermiogenesis, differences in the molecular mechanisms underlying spermatogenesis (*Murat et al., 2023*) or embryogenesis (*Lu et al., 2021*), or genetic redundancy (*Nowak et al., 1997*). Likewise, the lack of overlap with the cattle dataset might be attributed to differences in the protocol but also to the small number of MN peaks detected in the two cattle samples and the poor performance of the liftover from cattle to pig. The poor success of the liftover might have been caused by the double-step liftover from cattle to human and from human to pig

which was necessary due to the lack of a direct liftover between cattle and pig in the UCSC liftover tool.

A larger number of strongly enriched GO terms for the catalog of genes mapping nearby the MN when compared to the SN fractions was observed (Tables S4 and S5). This could be in part due to both, the low number of observed SN peaks when compared to MN sites, and the fact that the percentage of intergenic SN locations (57.5%) doubled that of intergenic MN peaks (28.6%). Hence, the GO analysis run for SN included a lower number of genes, and consequently, yielded a small number of enriched GO terms. The most enriched terms for the MN fraction included sensory perception, which is important for sperm homeostasis (*e.g.*, *SOD2*; (*Malivindi et al., 2018*)), capacitation (*e.g.*, *TRPV1*; (*Osycka-Salut et al., 2020*)) and chemotaxis guidance to the egg for fertilization (*e.g.*, *CA6*; (*Boué et al., 1995*)) and cell differentiation, development and morphogenesis, all with obvious links to embryogenesis (Table S4).

The search for TF motifs at MN peaks also indicated that MN peaks may carry instructions for embryo development. First, enriched motifs for several TFs related to embryo development were identified (Table S6). Some of these genes point toward a role of the paternal gamete in regulating embryo development. This is the case for *ATF4* (*Puscheck et al., 2015*) and *CHOP* (*Ali et al., 2018*), which are relevant in the stress response of the mouse embryo at key embryogenesis steps such as zygotic genome activation. FOXA1 is bound, in the chromatin of human sperm, to genomic regions corresponding to enhancer elements in embryonic stem cells and in several cell types of the embryo, which could suggest that the position of this TF in sperm may guide the location of relevant enhancers for embryo development (*Jung et al., 2019*). The identification of enriched Znf263 motifs is another relevant finding. *ZNF263* has been proposed as a master regulator of the genes showing paternal preferential expression in the early developing human embryo (*Leng et al., 2019*). The nearly significant enrichment (*Pvalue* = 0.052), of genes exhibiting paternal preferential expression in the human early embryo that correspond to pig orthologs mapping near a MN peak with predicted ZNF263 binding motif in sperm (Table S7) suggest that these MN peaks may be epigenetic marks in sperm that trigger the expression of specific genes upon fertilization. All in all, this data is in line with the hypothesis that nucleosome retention in sperm has guiding relevance in the early embryo development.

Although mature sperm is assumed to be transcriptionally silent, it carries a wide repertoire of RNAs related to spermatogenesis, fertilization, embryo development and offspring phenotype (reviewed in: (*Gòdia, Swanson & Krawetz, 2018*)), a large proportion of which were transcribed in transcriptionally active spermatogenic cells during previous stages of spermatogenesis. Our group has generated sperm RNA-Seq (*Gòdia et al., 2020a*; *Ablondi et al., 2021*; *Gòdia et al., 2020b*) and GWAS data for semen quality traits (*Gòdia et al., 2020b*) from porcine samples. The RNA-Seq and GWAS studies included 40 and 276 samples, respectively. Thirty-five of the 40 pigs that provided sperm for RNA-Seq also contributed data for the GWAS. The two boars used in the MNase study, also Pietrain, participated in the GWAS but not in the RNA-Seq. The regulation of transcription is modulated by nucleosome occupancy and the accessibility of the genome to the

transcriptional machinery (*Henikoff, Furuyama & Ahmad, 2004*). Our results show that the genes mapping within or nearby retained nucleosomes tend to display larger RNA abundance than the full set of genes annotated in the porcine genome (Fig. 3). These results also indicate that the genes that contributed detectable RNA levels by RNA-Seq in sperm, tend to map to nucleosome-retained loci in both the MN and SN fractions. Not only protein coding but also a family of regulatory RNAs, the piRNAs, were also significantly enriched at MN and SN peaks. Again, this data supports the notion that nucleosome occupancy is key in modulating gene expression. Moreover, the enriched co-location between the MN and SN sites and the catalog of piRNA regions (*Ablondi et al., 2021*) which abundance correlated with sperm quality phenotypes further suggests that this regulation during spermatogenesis may have phenotypic consequences on semen quality and reproduction. Thus, the nucleosomes and sub-nucleosomes associated to these RNAs may be leftovers from spermatogenesis and might provide useful information for the identification of markers of abnormal spermatogenesis and sperm quality.

In light of these results, and considering that nucleosome positioning can modulate gene expression (*Bai & Morozov, 2010*; *Lorch, LaPointe & Kornberg, 1987*; *Li, Carey & Workman, 2007*), it was hypothesized that sperm-retained nucleosomes could encompass DNA variants altering elements such as promoters or enhancers regulating the expression of genes that played a role during spermatogenesis. In such case, some degree of co-location between MNase peaks and GWAS hits should be observed. However, this trend between the MN and SN peaks with the GWAS hits for semen quality was not detected (Table 2 and Fig. S3) (*Gòdia et al., 2020b*).

The reasons, whether biological or technical, underlying the lack of a co-location trend with GWAS hits for semen quality and the catalog of sperm circRNAs cannot be determined. For the GWAS, one possibility is that because the GWAS hits are in fact, highlighting genomic regions in linkage disequilibrium with the causal functional variant, the GWAS region is expected to be much larger than this causal variant and probably most of the GWAS regions do not correspond to functional elements of the genome (*e.g.*, promoters, enhancers or genes) that could be overlapping with nucleosomes. Regarding the circRNAs, although the FC did not reach our threshold, none of the random permutations reached the same number of overlaps between MN peaks and the catalog of circRNAs indicating that, although modest, there might be a trend of co-location between circRNAs and MN peaks (Table 2).

Noteworthy, our results are not in disagreement with the hypothesis that sperm nucleosomes mostly map at gene-poor regions and repeat elements (*Carone et al., 2014*; *Samans et al., 2014*; *Oikawa et al., 2020*). The depletion of MN and enrichment of SN at LINEs and SINEs, respectively, might be indicating that nucleosomes in these repeat elements are more susceptible to MNase digestion as suggested by Carone and colleagues (*Carone et al., 2014*).

Four technical considerations in relation to the results of this study need to be taken into consideration. The first consideration is that the presence of somatic cells, which contain large number of histones in their chromatin, in the sperm samples could seriously mask the nucleosome profiles of the sperm cells. Confidence can be placed on the absence of

somatic cells in our samples for two reasons. First, commercial pig sperm doses rarely present somatic cell contamination. Second, the sperm samples were subjected to Bovipure™ purification which removes somatic cells. Moreover, the number of MN peaks identified in our study is close to the number of MN peaks found by *Hammoud et al. (2009)* in human, thereby indicating that the MNase profiles obtained in our study are not influenced by somatic cells. The second consideration is the fact that our chromatin extraction protocol yielded lower amount of DNA than what has been obtained in other species described in different studies, which could be indicating that our conditions were not fully optimized for pigs. The third consideration is that ChIP-Seq, either for H3 or for different histone modifications related to specific genome activity would have been more precise and informative than MNase-Seq. However, the amount of DNA obtained after the MNase digestion was limited and the immunoprecipitation would most likely have resulted in insufficient amounts of DNA for high throughput sequencing. Nevertheless, the MNase protocol has provided the required information to suggest that nucleosome retention in sperm may contain relevant information in relation to spermatogenesis, semen quality and embryo development. The fourth consideration relates to the fact that the results are based on a single pool of two samples and consequently, the robustness of the location of each particular nucleosomal site cannot be assessed. Nonetheless, the aim of this work was not to robustly map the specific location of nucleosomes but to evaluate whether the retained histones tend to co-map within functional features of the genome, which would indicate that nucleosome retention is not random but programmatic. The statistical confidence of our study was assessed within a sample, by testing the enrichment of positional overlap between the identified nucleosomal sites and the different functional features of the genome against the null basis of random location. Remarkably, all but two of the comparisons showed statistically significant co-location thereby indicating, that nucleosome retention in pig sperm is not stochastic and may owe to a reminiscent gametogenesis program and a future embryogenesis program. Still, since only one pool of two samples was evaluated, the results should be considered as preliminary and future research will be required to robustly establish the functional relevance of nucleosome retention in the sperm genome in swine.

## CONCLUSIONS

The results of our analyses show than the retained nucleosomes in the porcine sperm chromatin tend to overlap with different functional features of the genome such as promoters, exons, genes expressed in sperm and associated to semen quality, putative TFBS for TFs related to reproduction and embryogenesis. In conclusion, the findings suggest that nucleosome retention in mature spermatozoa is related to transcriptional regulation during spermatogenesis and is also an instructional contributor to early embryo development. Hence, interrogating the nucleosome occupancy in the sperm chromatin could help elucidating the biological basis of sperm quality and early embryo development and perhaps, could even assist in the search for biomarkers for these sets of traits.

## ACKNOWLEDGEMENTS

We would like to thank Sue Hammoud and all the members of Hammoud lab for their support. We also would like to thank Sam Balasch (Grup Gepork S.A., Catalonia) for supplying the sperm samples.

### Funding

This work was supported by the Spanish Ministry of Economy and Competitiveness (MINECO) under grant AGL2013-44978-R and grant AGL2017-86946-R and by the CERCA Programme/Generalitat de Catalunya. AGL2017-86946-R was also funded by the Spanish State Research Agency (AEI) and the European Regional Development Fund (ERDF). We had financial support from the Agency for Management of University and Research Grants (AGAUR) of the Generalitat de Catalunya (Grant Numbers 2014 SGR 1528 and 2017 SGR 1060) and from the Spanish Ministry of Science and Innovation, through the Severo Ochoa Programme for Centers of Excellence in R&D SEV-2015-0533 and CEX2019-000902-S. Marta Gòdia had a Ph.D. studentship from MINECO (Grant Number BES-2014-070560) and Short-Stay fellowships from MINECO at SSH group (EEBB-I-16-11528) and at CSIRO (EEBB-I-18-12860). The publication of this work was supported by the CSIC Open Access Publication Support Initiative through its Unit of Information Resources for Research (URICI). The funders had no role in study design, data collection and analysis, decision to publish, or preparation of the manuscript.

### Grant Disclosures

The following grant information was disclosed by the authors:
Spanish Ministry of Economy and Competitiveness (MINECO): AGL2013-44978-R, AGL2017-86946-R.
CERCA Programme/Generalitat de Catalunya.
Spanish State Research Agency (AEI) and the European Regional Development Fund (ERDF): AGL2017-86946-R.
Agency for Management of University and Research Grants (AGAUR) of the Generalitat de Catalunya: 2014 SGR 1528 and 2017 SGR 1060.
Spanish Ministry of Science and Innovation.
Severo Ochoa Programme for Centers of Excellence in R&D: SEV-2015-0533 and CEX2019-000902-S.
MINECO: BES-2014-070560.
Short-Stay fellowships from MINECO at SSH group: EEBB-I-16-11528.
CSIRO: EEBB-I-18-12860.
CSIC Open Access Publication Support Initiative through its Unit of Information Resources for Research (URICI).

### Competing Interests

The authors declare that they have no competing interests.

## Author Contributions

- Marta Gòdia conceived and designed the experiments, performed the experiments, analyzed the data, prepared figures and/or tables, authored or reviewed drafts of the article, and approved the final draft.
- Yu Lian analyzed the data, prepared figures and/or tables, authored or reviewed drafts of the article, and approved the final draft.
- Marina Naval-Sanchez conceived and designed the experiments, performed the experiments, analyzed the data, authored or reviewed drafts of the article, and approved the final draft.
- Inma Ponte performed the experiments, analyzed the data, authored or reviewed drafts of the article, and approved the final draft.
- Joan Enric Rodríguez-Gil performed the experiments, authored or reviewed drafts of the article, and approved the final draft.
- Armand Sanchez conceived and designed the experiments, authored or reviewed drafts of the article, and approved the final draft.
- Alex Clop conceived and designed the experiments, analyzed the data, prepared figures and/or tables, authored or reviewed drafts of the article, and approved the final draft.

## Data Availability

The data is available at the NCBI's Short Read Archive: SRR14117448, SRR14117447, SRR14117446.

The code is available at Figshare: Gòdia, M (2023): Commandlines_MNase_seq.sh. figshare. Online resource. https://doi.org/10.6084/m9.figshare.21997523.v1.

## Supplemental Information

Supplemental information for this article can be found online at http://dx.doi.org/10.7717/peerj.15520#supplemental-information.

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
