# Peer review of "Micrococcal nuclease sequencing of porcine sperm suggests enriched co-location between retained histones and genomic regions related to semen quality and early embryo development"

_PeerJ, doi:10.7717/peerj.15520_

## Round 0.1 · original submission · Major Revisions

All the reviewers suggested major changes in the manuscript before it gets published. Hence my recommendation is major revision.

Reviewer 1 ·

Basic reporting

- The article must be written in English and must use clear, unambiguous, technically correct text. The article must conform to professional standards of courtesy and expression. Use impersonal style, making use of passive voice when you talk about experimental processes or steps.

- The article should include sufficient introduction and background to demonstrate how the work fits into the broader field of knowledge. Relevant prior literature should be appropriately referenced.

Experimental design

- The submission should clearly define the research question, which must be relevant and meaningful. The knowledge gap being investigated should be identified, and statements should be made as to how the study contributes to filling that gap.

Validity of the findings

The conclusions should be appropriately stated, should be connected to the original question investigated, and should be limited to those supported by the results. In particular, claims of a causative relationship should be supported by a well-controlled experimental intervention. Correlation is not causation.

Additional comments

1. line 67-68 : check the style of references inside the text, in general, we used semicolon between reference
2. line 93-98 : in the end of introduction, it should write about objective and move the text from line 468-471 to the end of introduction
3. line 102-103 : Is the number of sample enough for statistical analysis?
4. line 113-125 : for abbreviation (PMSF, DTT, EGTA, MN, SN, TAE, etc.) , should give the full name first.
5. line 235-239 : numbers in parentheses are very confuse
6. Results : It is results or results and discussion, because many results were compared with previous study.
7. line 256-258 : should write in M&M
8. line 468-472 : move to the end of introduction

Reviewer 2 ·

Basic reporting

1 ‘Fig. 1a’ seems not correct in L164.
2 The authors should add more details in some key parts. For example, how do the authors calculate p-values in L266-L267?
3 What is the additional file 5 in L269?
4 The review of related work is not sufficiently thorough and not sufficiently specific. The authors should cite references if appropriate.

Experimental design

1 The samples are two ejaculates, each from a different healthy boar. I am wondering how the author evaluates the semen quality and replication of these samples. This may influence downstream analysis and their results a lot.

Validity of the findings

1 In today's research, open science is one of the essential aspects, and sharing the scripts is an important part. This way, future researchers can replicate your results when and where necessary. The authors should provide this information.

Reviewer 3 ·

Basic reporting

The manuscript can benefit from language editing. Especially the discussion section.
Background needs to be carefully introduced pointing out the ambiguities in previous histone retention findings. And how methodological choice might influence the results.
E.g. when it says cross-species comparison reveals positional conservation with human. It is absolutely crucial to first point out what findings in humans the authors are referring to.
It summarizes background based on relevant references, yet omits to mention the contradicting findings by Carone et al (Rando lab) on mouse sperm that question the mnase based methods employed by the labs cited in this article.
The structure conforms to Peer J standards
Figures support the statements and are well structured and labelled.
The Raw data have been submitted to a repository.

Experimental design

This is the first report on histone retention in porcine sperm, hence the findings are original.
Histone retention in mature mammalian sperm is a controversial topic with varying findings depending on the report, but potential importance for intergenerational information transmission and sperm quality. Hence it is a relevant topic that prompted further investigation.
The experiments and methods used are described in detail but also refer to previous descriptions which is not ideal. Also the code used to compute the data could be deposited which would facilitate an attempt of replication.
It is not justified why authors use mnase sequencing as opposed to histone Chip sequencing.
The purity of sample should be commented on and results provided. Even a small somatic contamination might result in artefacts given the low amounts of histones in sperm verus somatic cells.

Instead of commenting that the correlation of the 2 mnase treated samples was very high – it would be better to process them separately and show the reader how reproducible the two sperm samples are taking into consideration the respective input. But the authors acknowledge this caveot in the discussion.

Regarding motif enrichment analysis – what was used as a reference set of genomic loci?

Validity of the findings

The data mostly support the interpretations. Where a speculation the authors clearly say so.
I wonder why the authors report on TF signal in nucleosome sized peaks and not subnucleosome sized peaks and why they report plant Transcription factor motif enrichment.
Further from mouse and human studies CTCF is expected to be found in those regions.
Line 296 revise wording -> probably the word “significant” is missing.

The conclusion that human and pig peak results were similar – if 27% overlap even if this is tested against randomization – is misleading. Maybe a reinterpretation of ¼ similar and this is substantially more than expected by chance is more appropriate. Remains to be said that ¾ of the peaks do not overlap.

Furthermore the discussion lacks a mention of the so far inconclusive results regarding histone retention in sperm (Carone et al verus Erkek et al for instance).

The overlap of identified accessible regions and detected sperm RNA is interesting and could be more explicitly interpreted as those regions mirrowing prior transcription.
Have the authors also looked into immature porcine sperm RNA populations?


The discussion could be improved by removing some redundancy with the results section. (e.g line 341 the echoing of human sperm digestion size results)

It further reveals a detail that is not mentioned either in resuls or methods – this needs to be restructured:
“The extraction of nucleosomal DNA was less efficient than in experiments carried in the sperm
357 of other species (Hammoud et al., 2009) (Brykczynska et al., 2010) (Castillo et al., 2014)
358 (Samans et al., 2014) (Erkek et al., 2013), and it did not yield sufficient amount of DNA from the
359 electrophoretic bands for high throughput sequencing. Nevertheless, the total amount of digested
360 DNA was enough for sequencing and fragment size separation after read mapping using
361 bioinformatics tools. To directly sequence the MN and SN fractions, future experiments will
362 require to optimize the efficiency of the protocol or, less ideal, to process a larger number of
363 sperm cells to isolate sufficient DNA from each MNase electrophoretic band.”



I assume the following interpretation contains a mistake – or I might have misunderstood the statement due to unclear wording: “The nucleosome-associated DNAs covered ~ 0.3% of the porcine sperm, which is nearly ten-fold
365 and 75% more nucleosome retention than what has been described in human (Hammoud et al.,
366 2009) (Arpanahi et al., 2009) (Gatewood et al., 1987) (Brykczynska et al., 2010) (Castillo et al.,
367 2014) and mouse (Balhorn, Gledhill & Wyrobek, 1977) (Erkek et al., 2013) (Johnson et al.,
368 2016), respectively.”
Line 389 – describe the technical differences – also the cattle genome was not mentioned in the results section!

The interpretation of more enrichment could be an artefact of simply more input genes due to more MN peaks? Line 394

The discussion of ZNF263 again appears a bit redundant
Same for the RNA discussion.
Line 423 is unclear

Line 429 unclear “These results also
429 indicate that the genes present in sperm tend to map to nucleosome-retained loci in both the MN
430 and SN fractions.”

The overlap of retained regions with loci that correlate with semen quality is very interesting and could be explored more.

Line 441 and 442 simplify and revise: “DNA variants altering variants such as” – is very hard to read and the sentence is grammatically incorrect – remove “of”

Line 463: Might this rely on the “control” regions used in the distinct analysis

Additional comments

Abstract
- “However, its resolution could provide further light into the identification of the genomic
regions related to sperm biology and embryo development and it could also help
identifying molecular markers for sperm quality and fertility traits.”
Please revise this sentence in terms of language and contents. E.g. Describe what is meant my sperm biology and what is meant by provide light into the identification
- What is meant by functional elements?
- DNA is singular
“we found enriched co-occupancyof the RNAs present in pig sperm and the RNAs related to sperm quality, with the mononucleosomal Peaks.” Cooccupancy is an interpretation – I advice to better refer to “signal”
- Please define “sperm biology”?
- Line 81/82 “Nucleosome positioning in the genome and chromatin accessibility are critical in the regulation of gene expression and have been linked to multiple phenotypes (Lai & Pugh, 2017).” I assume the authors mean altered nucleosome positioning has been linked to phenotypes?
- Line 87: I am not sure whether sperm biology and embryo development can be termed “traits”

---

## Round 0.2 · Minor Revisions

All the reviewers suggested minor changes in the manuscript before it gets published. Hence my recommendation is minor revision.

Reviewer 1 ·

Basic reporting

1. The article must be write in passive sentence such as line 41, 48 , 54 , 58, and 60 also in introduction, materials and methods, results and discussion.

Experimental design

For the sample collection, are two samples enough for validation or statistical analysis? How many replication should define? For my opinion, more than 3 samples will be more reliable.

Validity of the findings

Replication studies will be considered.

Additional comments

Line 62-63 : please rewrite to give more information about the relationship
Line 63 : remove "spermatogenesis, sperm quality, fertility)
Line 64 : remove "pilot" (if this is pilot study, the article should change to short communication article)
Line 72,78 : change fertility to fertilization
Line 81 : 5-15 % of DNA remains or 5-15% of human had DNA remains
Line 108-110 : please check to format to writing "Jung et al. ....... (Jung et al., 2017)"
Line 119-120 : Change to passive sentence (if you write in this style, it look you tend to be self-citation.
Line 621 : From the results showed more than
Line 624 : change "In conclusion, we can speculate" to "It will be concluded"
Line 632 : We would like to thank

Reviewer 2 ·

Basic reporting

My concerns are well addressed by the authors.

Experimental design

My concerns are well addressed by the authors.

Validity of the findings

My concerns are well addressed by the authors.

Reviewer 3 ·

Basic reporting

Some remaining comments with regards to clarity of the wording:

Tiny error in the new sentence:
“Nevertheless, we MNase protocol has provided the required information to suggest that nucleosome retention in sperm may contain relevant information in relation to spermatogenesis, semen quality and embryo development»

Also when I commented on : cross-species comparison reveals positional conservation with human.» It is absolutely crucial to first point out what findings in humans the authors are referring to.
I did not think of the addition of a citation but rather in light of the ambiguous finding from distinct studies it would be important to mention to what “party” the human studies referred to belongs A) those that find enrichment in functional elements or B) those that find enrichment in gene deserts!

Point 23: I think it would be important to indicate that default parameters were used – which for Homer menas that to the input comparable genomic regions will be randomly picked while considering an equvivalent GC content.

Also I still struggle a lot to understand the new sentence.
“The boars used for RNA-Seq were also included in the GWAS. The 2 boars used in the MNase study, were also Pietrain and were included in the GWAS but not in the RNA-Seq.” _
It is the samples from boars that are used – but I don’t understand what you mean by were included in the GWAS. Maybe GWAS analysi used 4 samples: 2 from the same boars that yielded samples for RNA seq and 2 from the same boars that yielded samples for MNase seq.
I really had to read this 5 Times until I understood what is meant – or it is still not what you meant?

Revise wording:
“These results also indicate that the genes that show RNA levels in sperm tend to map to nucleosome-retained loci in both the MN and SN fractions.” _
Also here this is still far from an elegant wording! Try: “the genes that sperm RNA map to tend to coincide nucleosome retained loci in both……..

I am well aware how the term functional elements is used in genomics – yet depending on how this is exactly defined the loci it refers to differ hence I was asking for a specification here, which now you provided.  if I read functional elements for instance I think also of enhancers – which here were not investigated.

Experimental design

The authors provided an argument on why they didn't analyse the 2 bull samples separately.

Validity of the findings

It is clearly indicated that this is a qualitative not a fully quantitative analysis. The authors did a really great job in addressing all concerns. I really enjoyed reading the new version of this manuscript!

Additional comments

The authors did a really great job in addressing all concerns. I enjoyed reading the new version of this manuscript!

---

## Round 0.3 · accepted · Accept

The original Academic Editor is not available and so I have taken over the handling of your submission. In my opinion, the manuscript is ready for publication.